# A Low-Molecular-Weight Polyethylenimine/pDNA-VEGF Polyplex System Constructed in a One-Pot Manner for Hindlimb Ischemia Therapy

**DOI:** 10.3390/pharmaceutics11040171

**Published:** 2019-04-08

**Authors:** Xiaoshuang Guo, Zihan Yuan, Yang Xu, Xiaotian Zhao, Zhiwei Fang, Wei-En Yuan

**Affiliations:** Engineering Research Center of Cell & Therapeutic Antibody, Ministry of Education, and School of Pharmacy, Shanghai Jiao Tong University, Shanghai 200240, China; guoxs1012@163.com (X.G.); 15216777551@163.com (Z.Y.); xuyang821@sjtu.edu.cn (Y.X.); zhao_xiaot@hotmail.com (X.Z.); 13236095968@163.com (Z.F.)

**Keywords:** peripheral arterial disease, gene therapy, interpenetrating network, VEGF, hindlimb ischemia therapy

## Abstract

Peripheral arterial disease (PAD) is often characterized by continued reduction in blood flow supply to limbs. Advanced therapeutic strategies like gene therapy could potentially be applied to limb ischemia therapy. However, developing a gene delivery system with low toxicity and high efficiency remains a great challenge. In this study, a one-pot construction was used to integrate vector synthesis and polyplex fabrication simultaneously in a simple and robust manner. We fabricated an interpenetrating gene delivery network through the physical interaction between low-molecular-weight polyethylenimine (PEI 1.8 kDa) and plasmid DNA (pDNA) and the chemical bonding between PEI and glutaraldehyde (GA), which was named the glutaraldehydelinked-branched PEI (GPEI) polyplex. The final GPEI polyplex system was pH-responsive and biodegradable due to the imine linkage and it could successfully deliver desired vascular endothelial growth factor (VEGF) pDNA. Compared with PEI (25 kDa)/pDNA polyplexes, GPEI polyplexes showed lower cytotoxicity and higher transfection efficiency both in vitro and in vivo. In addition, we demonstrated that GPEI polyplexes could efficiently promote the formation of new capillaries in vivo, which may provide a practicable strategy for clinical hindlimb ischemia therapy in the future.

## 1. Introduction

Characterized by continued reduction in blood flow to one or more limbs, peripheral arterial disease (PAD) is mainly caused by atherosclerosis, and it can cause severe ischemic limb disease [1]. The prevalence of PAD has been gradually increasing in recent years due to the changes in dietary structure, aging of the population, and the growing frequency of vascular surgery. For example, the average risk of developing arteriosclerosis for diabetic patients is 19 times higher than that of healthy individuals, and these diabetic patients are more likely to suffer from PAD [2,3]. Besides, thromboangiitis obliterans and other chronic diseases can induce severe PAD [4,5]. Clinically, current therapeutic approaches to PAD include antiplatelet therapy (aspirin and clopidogrel) [6], anticoagulants (cilostazol) [7], statin therapy [8], and vascular dilation drugs (Anplag) [9] as well as cell-based therapy [10]. Even with such multiple choices, existing clinical strategies are still fraught with some challenges like low therapy efficacy and poor prognosis. Gene therapy may provide an advanced therapeutic strategy for PAD, especially chronic lower extremity ischemia, via the delivery of nucleic acid-based drugs (DNA or RNA) to target cells [11]. It was reported in a preclinical study that this strategy could promote new blood vessel formation and improve blood supply to ischemic lower limbs [11]. Target cells that are transfected successfully can translate gene drugs into cytokines, contributing to therapeutic angiogenesis [12].

To achieve therapeutic angiogenesis, researchers have focused on gene therapy with proangiogenic factors such as vascular endothelial growth factor (VEGF), hepatocyte growth factor (HGF), and fibroblast growth factor (FGF) [13]. Among them, VEGF can promote the growth of vascular endothelial cells (ECs), and it has been confirmed that VEGF induces a remarkable angiogenic response in various in vivo models [14].

Safe and efficient gene vectors have been extensively studied since naked DNA can be degraded by nucleases easily. Gene vectors include viral vectors and non-viral vectors. Nowadays, due to the fact that potential immunogenicity and antigenicity of viral vectors can lead to a series of severe safety issues, non-viral vectors have been widely studied for their high safety. In addition, non-viral vectors can be chemically modified in structure easily [15]. 

Polyethyleneimine (PEI) is one of the most classic non-viral vectors, and PEI with molecular weight of 25 kDa, considered as the gold standard of non-viral vectors, has a strong ability to condense and deliver nucleic acids [16]. However, the high molecular weight of PEI 25 kDa leads to high cytotoxicity since the polymer cannot be degraded and metabolized in vivo [17]. In contrast, low-molecular-weight PEI such as PEI 1.8 kDa has low cytotoxicity, but its transfection efficiency is relatively low [17]. Currently, in response to these problems, a common strategy is to crosslink low-molecular-weight PEI with biodegradable linkers [18]. Not only does this strategy enhance the transfection efficiency of low-molecular-weight PEI, but also it reduces the toxicity of polymers [19,20]. However, the complicated design of these polymers requires demanding chemistry synthesis skills and long purification process, often limiting practical use.

Thus, our group designed an integrative plasmid DNA (pDNA) delivery system with glutaraldehydelinked-branched PEI (GPEI) polyplexes (Figure 1) [21]. Glutaraldehyde (GA) was utilized as cross-linking agent to crosslink low-molecular-weight PEI (PEI 1.8 kDa). A one-pot construction was used to simultaneously achieve the integration of vector synthesis and polyplex preparation in a simple step. We mixed VEGF pDNA, PEI 1.8 kDa, and GA solution together in the aqueous environment. Throughout this robust system, PEI 1.8 kDa could easily and efficiently condense pDNA via electrostatic interaction to form polyplexes. Then, the aldehyde groups in GA could further react with the amine groups of PEI 1.8 kDa to form imine linkages, which could condense the pDNA again in theory. Ideally, the electrostatic interaction between PEI and pDNA and the chemical bonding between PEI and GA could finally come into an interpenetrating network, resulting in high stability and dense structure. Our experimental results showed that polyplexes prepared by this twice-condensation method showed smaller particle sizes and higher transfection efficiency. The stability of polyplexes system was enhanced, which could effectively reduce nucleic acid leakage. Besides, the preparation of GPEI polyplexes described here was quite simple, robust, and practicable. It avoided time-consuming chemical synthesis and purification as well as the use of organic solvents in the biomaterial system, resulting in time savings and desired biocompatibility.

The chemical characterization and degradation experiments of GPEI analog GAPEI have been reported in our previous work [21]. In the study here, we investigated the physicochemical properties and pDNA condensing ability of GPEI. Then, we studied in vitro cytotoxicity and transfection efficiency of GPEI polyplexes in human umbilical vein endothelial cells (HUVECs) to confirm the one-pot system constructed here was with both high biocompatibility and efficacy. Finally, the in vivo delivery efficiency and treatment effect of GPEI polyplexes were evaluated in the well-established rat model of hindlimb ischemia.

## 2. Materials and Methods

### 2.1. Materials

Glutaraldehyde (GA) and branched polyethylenimine (1.8 and 25 kDa) were purchased from Sigma-Aldrich. VEGF-A (1239 bp) pDNA encoding GFP was constructed by Bioroot Biology (Shanghai, China). 

### 2.2. Cell Culture

Human umbilical vein endothelial cells (HUVECs, primary cells) were purchased from ALLCELLS (Shanghai, China) and cultured in RPMI-1640 (GIBCO BRL, Grand Island, NY, USA) containing 10% fetal bovine serum (FBS) (GIBCO BRL, Grand Island, NY, USA) and 1% antibiotic (GIBCO, Grand Island, NY, USA) at 37 °C with 5% CO_2_ in the cell incubator (Thermo Fisher Scientific, Waltham, MA, USA).

### 2.3. Preparation of GPEI Polyplexes

According to the formula showing in Table 1, GPEI polyplexes were prepared at various PEI 1.8 kDa/pDNA weight/weight (*w*/*w*) ratios. First, pDNA solution (20 ng/μL) was mixed with PEI 1.8 kDa solution (2 mg/mL) and H_2_O. These mixtures were incubated at room temperature for 30 min (first condensation). Then, GA solution (0.125%) was added and these mixtures were heated in a water bath at 65 °C with oscillation for 30 min (second condensation). Finally, the prepared GPEI polyplex solution was stored at 4 °C for further use.

### 2.4. Characterization of Polyplexes

#### 2.4.1. Agarose Gel Electrophoresis (AGE)

AGE was used to evaluate the condensation ability of GPEI. GPEI polyplexes were prepared at various *w*/*w* ratios (from 0.01 to 0.5). Then, 1 μL of loading buffer (30 mM ethylenediaminetetraacetic acid (EDTA), 36% (*v*/*v*) glycerol, 0.05% (*w*/*v*) xylene cyanol, and 0.05% (*w*/*v*) Bromophenol Blue) was added to 5 μL of marker. Then, 2 μL loading buffer were added into 10 μL of GPEI polyplexes at various *w*/*w* ratios respectively. PEI 25 kDa polyplexes (*w*/*w* ratio = 2) were used as the positive control and naked pDNA (40 ng/μL) was used as the negative control. All the samples were loaded on 1.0% agarose gel in Tris-Acetate-EDTA (TAE) running buffer and electrophoresed for 40 min at 120 V. The retardation of pDNA was imaged with Tanon-3500 Gel Imaging System.

#### 2.4.2. Particle Size, Zeta Potential and Morphology Measurements

The particle size and zeta potential of GPEI polyplexes at various *w*/*w* ratios were measured by Particle Size Analyzer (Brookhaven Particle Size Analyzer 90 Plus, Brookhaven Instruments Corporation, Holtsville, NY, USA), and PEI 25 kDa polyplexes (*w*/*w* ratio = 2) were used as the positive control. The morphology of GPEI polyplexes at *w*/*w* ratio of 5 was examined by transmission electron microscopic (TEM) (Tecnai G2 spirit Biotwin, Thermo Fisher Scientific, Waltham, MA, USA).

### 2.5. In Vitro Cytotoxicity

In vitro cytotoxicity of GPEI polyplexes was measured by using Cell Counting Kit-8 (CCK-8) (DOJINDO LABORATORISE, Shanghai, China) reagent. First, HUVECs were seeded into 96-well plates (1 × 10^4^ cells/well) and cultured in cell incubator for about 24 h. Next, the medium was replaced by 50 μL new culture medium (without FBS), and 10 μL GPEI polyplex solutions at various *w*/*w* ratios (from 1 to 20) were added into wells for 4-h and 24-h incubation. Each ratio was repeated 6 times. Then, each well was replaced by 10 μL of CCK-8 reagent and 50 μL of new culture medium. About 2 h later, the absorbance was measured by multifunctional microplate reader (SpectraMax M3 Multi-Mode Microplate Reader, Sunnyvale, CA, USA). PEI 25 kDa polyplexes at the same *w*/*w* ratios were used as the control.

### 2.6. Intracellular Uptake

In order to have a direct observation of the intracellular uptake of GPEI polyplexes, the intracellular localization of GPEI polyplexes in HUVECs was measured by a Super-Resolution Multiphoton Confocal Microscope (SMCM) (TCS SP8 STED 3X, Lecia, Germany) [22]. GPEI polyplex solutions (*w*/*w* ratio = 5, 10) were prepared as regular methods and the pDNA was labeled with the fluorophore Cy3 (GenePharma, Shanghai, China). HUVECs were seeded into 12-well plates (5 × 10^5^ cells/well) with cell slides (WHB Scientific, Shanghai, China) and cultured for about 24 h. Then, the medium was replaced by 1 mL new culture medium (without FBS) and 200 μL GPEI polyplex solutions. After 2 h, cells were washed five times with phosphate buffer saline (PBS) and 2 mL LysoTracker Green (100 nM) (Beyotime, Shanghai, China) were added to mark the lysosome for 50 min at 37 °C. Next, cells were washed five times with PBS and 1 mL trypan blue solution (0.4%) (Yeasen Biotech, Shanghai, China) was added to mark viable cells for 2 min at room temperature. Then, cells were washed five times with PBS and 1 mL 4% paraformaldehyde was added to fix cells for 30 min at room temperature. Finally, cells were washed five times with PBS and 2 mL DAPI (1 μg/mL) (Roche Diagnostics, Mannheim, Germany) was added to mark the cell nucleus for 4 min at 37 °C and investigated by SMCM. Similarly, the 4-h intracellular uptake was conducted by the same method.

### 2.7. In Vitro Cell Transfection

HUVECs were seeded into 24-well plates ((5~10) × 10^4^ cells/well) and cultured in cell incubator for about 24 h. Then, the medium was replaced by 500 μL new culture medium (without FBS) and 100 μL GPEI polyplex solutions at various *w*/*w* ratios (from 1 to 20) were added into wells for 4 h incubation, and each ratio was repeated for 3 times. PEI 25 kDa polyplexes (*w*/*w* ratio = 2) were used as the positive control, naked pDNA as the negative control, and PBS as the blank control. Then, each well was replaced by 1 mL new culture medium. Seventy-two hours later, flow cytometry (FCM) (BD LSRFortessa, Franklin, NJ, USA) was utilized to determine the percentage of cells with green fluorescence.

### 2.8. Hindlimb Ischemia Model Study

All animal experiments were conducted strictly in accordance with the guidelines approved by the Regulations for the Administration of Affair Concerning Laboratory Animals for Shanghai Jiao Tong University, the National Institutes of Health Guide for Care and Use of Laboratory Animals (GB14925-2010), the Regulations for the Administration of Affairs Concerning Experimental Animals (China, 2014). The Animal Ethics Committee of Shanghai Jiao Tong University reviewed and approved the experiments (SJTU, No. A2017073, approval date: 23 December 2017).

Sixteen healthy Sprague–Dawley rats (male, weight 150–200 g) were raised under standard laboratory conditions in the laboratory animal facility in the school of pharmacy, Shanghai Jiao Tong University. One week later, rats weighing about 250 g were intraperitoneally injected with pentobarbital sodium (50 mg/kg) for anesthesia, and operations were performed according to reported article as well as the instructional video [23]. After the operation, rats were randomly divided into four groups (*n* = 4). When the skin wound at surgical sites was completely healed, drugs (0.1 mL/100 g) were injected intramuscularly at the model site once every three days. The blank control group was injected with normal saline. The negative control group was injected with VEGF pDNA (100 ng/μL). The positive control group was injected with PEI 25 kDa polyplexes (*w*/*w* ratio = 2) containing VEGF pDNA (100 ng/μL). And the GPEI group was injected with GPEI polyplexes (*w*/*w* ratio = 5) containing VEGF pDNA (100 ng/μL). In addition, each rat was injected with 5-Bromo-2-deoxyuridine (BrdU) (Sigma-Aldrich, St. Louis, MO, USA) once every three days [24]. After four weeks of treatment, rats were sacrificed for further study.

### 2.9. In Vivo Cytotoxicity

In vivo cytotoxicity of GPEI polyplexes was evaluated by hematoxylin–eosin (HE) staining. When rats were sacrificed, the main organs (including heart, liver, spleen, lungs and kidneys) of rats were immediately harvested and fixed in 4% paraformaldehyde, sectioned, and stained with HE. After staining, the glass slides were observed under optical microscope.

### 2.10. In Vivo Therapeutic Effects

Gastrocnemii at the surgical site of sacrificed rats were collected and fixed in 4% paraformaldehyde and assessed by HE staining. Angiogenesis was assessed by both CD34 immunofluorescence staining [25] and CD31 immunohistochemical staining [26], and the capillary density of gastrocnemii was analyzed semi-quantitatively by software ImageJ. 5-Bromo-2-deoxyUridine (BrdU) antibody was used to label proliferating cells in the tissue. Besides that, part of gastrocnemii was immediately frozen at −80 °C after the sacrifice, which were then used to quantitatively evaluate the expression of VEGF protein in the tissue by Western blotting (WB).

### 2.11. Statistical Analysis

Data of each independence measurement were presented as mean values ± standard deviation (S.D.). Statistical analysis was tested by independent sample *t*-test, and statistical significance was assigned as *: *p* < 0.05, **: *p* < 0.01, or ***: *p* < 0.001.

## 3. Results and Discussion

### 3.1. Characterization of Polyplexes

#### 3.1.1. Agarose Gel Electrophoresis (AGE)

The ability of GPEI to condense pDNA into nanoparticles was measured by AGE. As shown in Figure 2, when the *w*/*w* ratio was over 0.03:1, there was no pDNA electrophoretic band present, which indicated that GPEI had a strong ability to condense nucleic acids efficiently, even at a small *w*/*w* ratio.

#### 3.1.2. Particle Size, Zeta Potential, and Morphology Measurements

The particle size and zeta potential of the nanoparticles have a great influence on the delivery efficiency of GPEI polyplexes. As shown in Figure 3A, particle sizes of GPEI polyplexes at various *w*/*w* ratios were in the range of 200 to 300 nm, which were significantly lower than that of PEI 1.8 kDa. And the polydispersity index (PDI) of GPEI polyplexes was low (Figure 3B), confirming that particle sizes of GPEI polyplexes in the system was uniform. We mixed VEGF pDNA, PEI 1.8 kDa, and GA aqueous solution together in a one-pot manner to prepare GPEI polyplexes. The PDI of some groups exceeded 0.2, which may be caused by uneven cross-linking during preparation. Excessive and uneven cross-linking of glutaraldehyde and amine groups may cause multiple polyplexes to link together, contributing to heterogeneity in particle size. Therefore, polyplexes should be oscillated continuously during heating to ensure a narrow range of particle size. The zeta potential of GPEI polyplexes ranged from 30 to 50 mV, which was higher than that of PEI 1.8 kDa (Figure 3C). In addition, TEM images also directly demonstrated that particle sizes were around 150 nm at the *w*/*w* ratio of 5 (Figure 3D). All above data showed that the GPEI polyplexes formed by twice-condensation of PEI 1.8 kDa with GA can efficiently enhance the condensation capacity of PEI 1.8 kDa, as well as reduce the particle sizes of polyplexes.

### 3.2. In Vitro Cytotoxicity

Safety and biocompatibility are important indicators for evaluating the practicability of applying polycation in biological systems. Thus, in vitro cytotoxicity of GPEI polyplexes was measured by CCK-8 reagent in this study. As shown in Figure 4, the cytotoxicity of GPEI polyplexes was significantly lower than that of PEI 25 kDa polyplexes. GPEI showed no obvious cytotoxicity when at low *w*/*w* ratios such as 1, 3, 5, and 10. The cytotoxicity of GPEI was higher with increasing *w*/*w* ratios, and the cell viability of GPEI was only 50% when the *w*/*w* ratio reached 30. From the results we could also find that there were fewer cells alive in 24-h cytotoxicity test compared with that in 4-h cytotoxicity test. The cell viability of some groups exceeded 100%, probably because the cytotoxicity of polyplexes with small w/w ratios was low and cells were still proliferating during the basal medium incubation. One reason accounting for the low cytotoxicity of GPEI in this study is that GPEI here is prepared by conjugating the aldehyde groups in GA with amine groups in PEI (1.8 kDa) to generate imine linkage, which is pH-responsive and could degrade under the acidic environment of endosomes.

### 3.3. Intracellular Uptake

Intracellular uptake is a critical step in the delivery of pDNA by GPEI. In this study, we performed intracellular localization of GPEI polyplexes to capture the intracellular uptake of GPEI polyplexes in HUVECs. The pDNA used in GPEI polyplexes was labeled with Cy3, which could emit red fluorescence under excitation. Therefore, the red fluorescent signals indicated the existence of pDNA, which is also the representation of the GPEI polyplex position, while the green fluorescent signals in the system suggested the positions of endosomes. As shown in Figure 5, a few GPEI polyplexes were endocytosed into cytoplasm after 2 h and many polyplexes were encapsulated in endosomes after 4 h. These results indicated that GPEI polyplexes could be endocytosed into cells and deliver pDNA successfully. In addition, some red dots appeared in the nuclear area (Figure 5), indicating that some of GPEI polyplexes could escape from endosomes and readily enter into nucleus for gene expression.

### 3.4. In Vitro Cell Transfection

Transfection efficiency is an important aspect when evaluating the performance of a gene delivery system. As shown in Figure 6, the transfection efficiency of GPEI group become higher with the increasing *w*/*w* ratios (from 1 to 20). Compared to that in PEI 25 kDa group (*w*/*w* ratio = 2), the transfection efficiency in GPEI group was higher. Significant improvement in transfection efficiency could be seen in GPEI group, when compared to that in naked pDNA group. No fluorescent signal was detected in the naked DNA group, indicating that naked DNA could hardly enter cells and been expressed, therefore, a safe and efficient delivery vector is required.

### 3.5. In Vivo Cytotoxicity

Main organs including heart, liver, spleen, lungs, and kidneys stained with HE were imaged by optical microscope to evaluate long-term toxicity. As shown in Figure 7, the staining results in all groups were comparable with each other, which means there is no significant difference among different groups. This may be explained by two reasons. Firstly, polyplexes in this study were injected intramuscularly rather than intravenously at the model site, and therefore they may not circulate around the body and cause the obvious toxicity in various organs. Secondly, the injected polyplexes are macromolecules that cannot be easily absorbed and distributed after local injection. They may be first degraded into small molecules and then absorbed and distributed, but the metabolized small molecules during this stage have low toxicity and thus cannot cause significant organ toxicity.

### 3.6. In Vivo Therapeutic Effects

According to the experimental results in vitro, the *w*/*w* ratio of 5 and 10 might be the ideal formulations for research in vivo, considering the cytotoxicity and transfection efficiency. Although the GPEI 5:1 showed a lower transfection efficiency, its cell viability was higher than that of GPEI 10:1. Therefore, we chose GPEI 5:1 for in vivo animal model study so that animals would face fewer safety risks.

In order to evaluate the in vivo therapeutic effect of the GPEI polyplexes, hindlimb ischemia model and corresponding evaluation methods were established. As shown in Figure 8A, there were visible gaps in the muscle fiber of blank group and naked VEGF group. In contrast, PEI 25 kDa 2:1 group and GPEI 5:1 group showed denser muscle fibers and presented no symptoms of muscular atrophy.

BrdU is a thymidine analog that can incorporate into the DNA molecule being replicated during the DNA synthesis phase (S phase) instead of thymine. Therefore, the cell proliferation of gastrocnemii can be analyzed by detecting the fluorescence signal of BrdU. As shown in Figure 8B, the signal density in each group was not significantly different, implying that the proliferation of cells in each group was comparable.

CD34 immunofluorescence staining and CD31 immunohistochemical staining were used for morphological observation to evaluate neovascularization in gastrocnemii. In Figure 8C, although some CD34 signals could be detected in the blank group and naked VEGF group, almost no intact vascular morphology was observed in gastrocnemii. In contrast, a greater number of areas of neovascularization were clearly observed in the PEI25 kDa 2:1 group and the GPEI 5:1 group. And results in Figure 8D also showed that the PEI25 kDa 2:1 group and GPEI 5:1 group showed more newborn capillaries. The capillary density of gastrocnemii was then analyzed semi-quantitatively (Figure 8E,F). The result was consistent with the morphological observation and confirmed our expectation that the gene therapy of VEGF could promote capillary angiogenesis.

The VEGF protein expression in gastrocnemii in vivo was determined by WB and GAPDH served as the loading control. As shown in Figure 8G,H, VEGF protein levels in two polyplex groups were both significantly higher than that in other two groups, indicating that VEGF pDNA in the polyplex groups was successfully delivered, transcribed, translated, and finally released.

All experimental results above demonstrated that GPEI polyplexes could deliver VEGF pDNA into cells efficiently and increase the expression level of VEGF protein in vivo. According to our previous work [19,20,27,28], plasmid DNA encoding anti-VEGF-shRNA or anti-VEGF-siRNA [29] could be delivered by polyplexes to inhibit the expression of VEGF, which could further limit the growth of tumors. Therefore, we could conclude that the increased VEGF expression and capillary formation demonstrated in this study come from the high translation of delivered VEGF pDNA rather than simple upregulated expression induced by injury. As reported, VEGF protein could promote the formation of new capillaries, thereby improving the blood supply of the ischemic hind-limbs, increasing the nutrient transport of the ischemic hindlimbs, and even alleviating muscle atrophy to some extent. Therefore, we believe the GPEI/pDNA-VEGF polyplex system developed in this study could provide a practicable strategy for future clinical hindlimb ischemia therapy.

## 4. Conclusions

In order to achieve effective treatment of PAD, researchers have focused on gene therapy with angiogenic factors [30]. As the most classic non-viral polycation carrier, PEI has been widely studied in recent years [31]. Many scientists have put a lot of effort to reducing the toxicity of PEI, with methods like modifying PEI with lipids [32], polyethylene glycol (PEG) [33,34], and fluorine [35,36], and crosslinking low-molecular-weight PEI with biodegradable chemical bonds [18,19,20]. However, all these methods complicate the system and limit practical use since intricate synthesis, purification, and the use of organic reagents are often involved. In contrast, the preparation of GPEI polyplexes in our one-pot manner was simple, practicable, and convenient. It avoided long-term synthesis and purification and the use of organic solvents, and saved lots of time. However, the GPEI polyplex presented here still lacks targeting groups, thus it can only be injected locally at the ischemic site, which needs further improvement.

In summary, the novel gene vector GPEI has been developed by twice-condensation in our study. Compared with other existing polycation vectors, advantages of GPEI are as follows: (1) the preparation method of GPEI polyplexes is quite simple, robust, and practicable, allowing researchers to reproduce the delivery system successfully; (2) the reaction was fast and no organic solvents were used in this biomaterial system, avoiding long-term synthesis and time-consuming purification; (3) low-molecular-weight PEI (1.8 kDa) was condensed twice to form the interpenetrating network, enhancing the stability of polyplexes and reducing nucleic acid leakage effectively; (4) the imine linkage conjugated in the polymer could respond to acid environment, promoting the release of nucleic acid intracellularly; and (5) GPEI polyplexes linked with imine bonds are biodegradable, leading to low cytotoxicity.

In our constructed polyplex delivery system, GPEI can condense nucleic acid into nanoparticles effectively with homogeneous particle size and appropriate zeta potential and the nucleic acid can be successfully delivered and released. GPEI polyplexes show low cytotoxicity and high transfection efficiency both in vitro and in vivo. Therefore, we believe the GPEI/pDNA-VEGF polyplex system described here could be a promising candidate for the future hindlimb ischemia therapy.

## Figures and Tables

**Figure 1 pharmaceutics-11-00171-f001:**
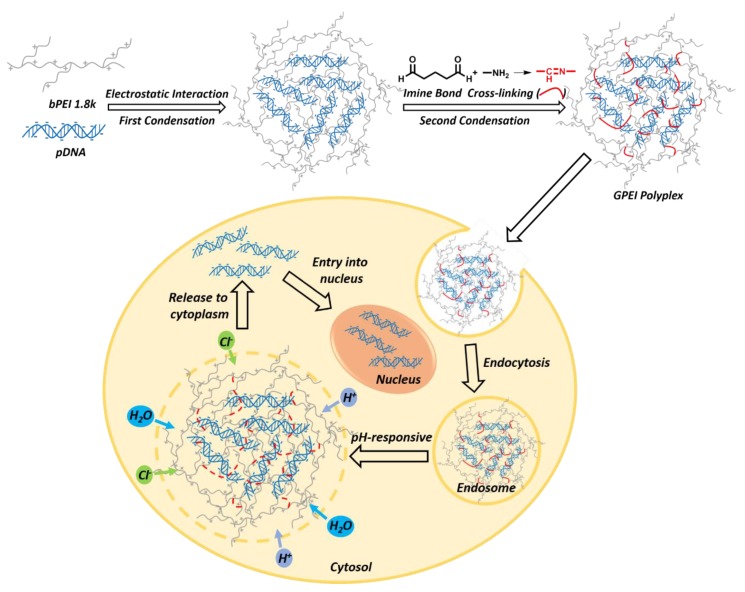
Schematic illustration showing the construction and delivered process of glutaraldehydelinked-branched PEI (GPEI) polyplex.

**Figure 2 pharmaceutics-11-00171-f002:**
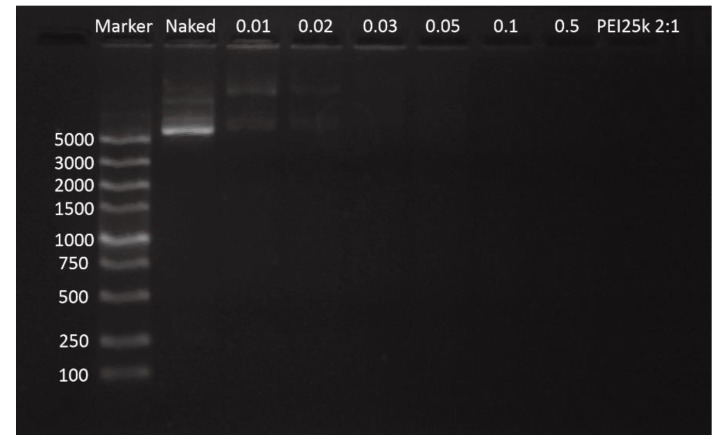
Agarose gel electrophoresis of the GPEI-pDNA polyplexes.

**Figure 3 pharmaceutics-11-00171-f003:**
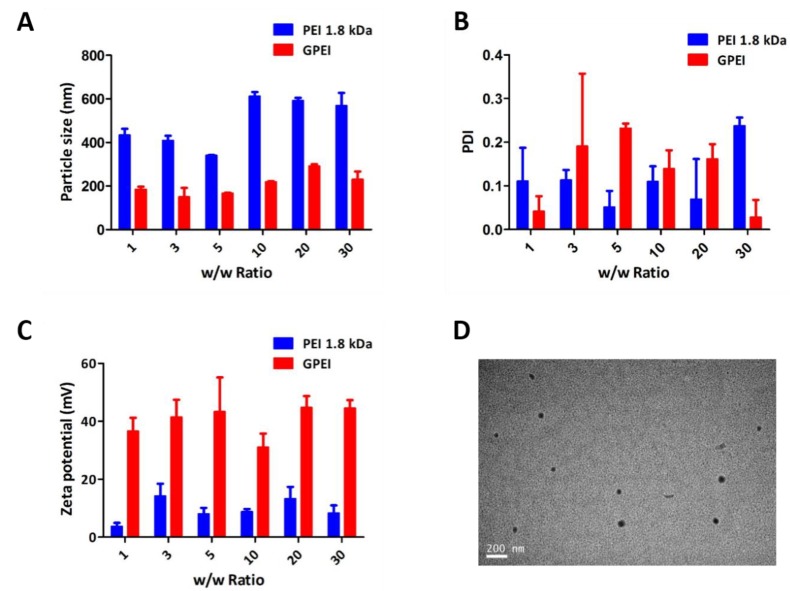
Physiochemical properties of the GPEI polyplexes. (**A**) Particle sizes, (**B**) polydispersity index (PDI), and (**C**) zeta potential of the GPEI and PEI 1.8 kDa polyplexes. (**D**) TEM images of the GPEI polyplexes (*w*/*w* ratio = 5). Bars = 200 nm.

**Figure 4 pharmaceutics-11-00171-f004:**
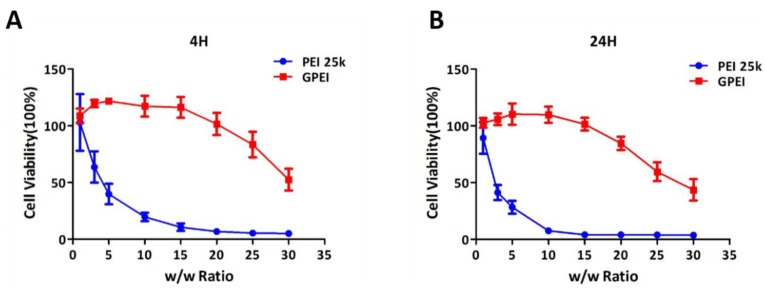
Cell viability of human umbilical vein endothelial cells (HUVECs) treated with the GPEI and PEI 25 kDa polyplexes. (**A**) Cell viability of HUVECs for 4 h. (**B**) Cell viability of HUVECs for 24 h. Data are shown as mean ± S.D. (*n* = 6).

**Figure 5 pharmaceutics-11-00171-f005:**
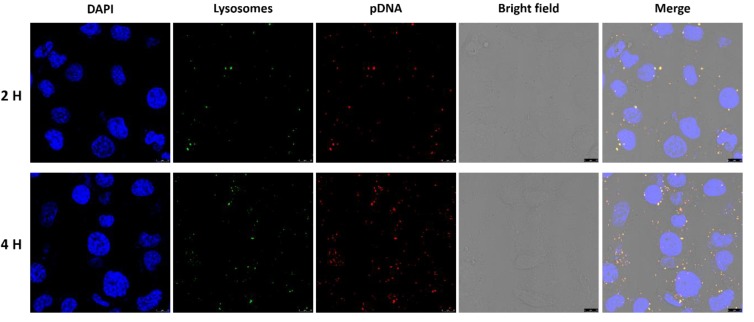
Images of intracellular uptake tests. HUVECs were treated with GPEI polyplexes (*w*/*w* ratio = 5) for 2 and 4 h. Bars = 10 μm. Blue: DAPI; green: LysoTracker Green; red: Cy3-labeled pDNA.

**Figure 6 pharmaceutics-11-00171-f006:**
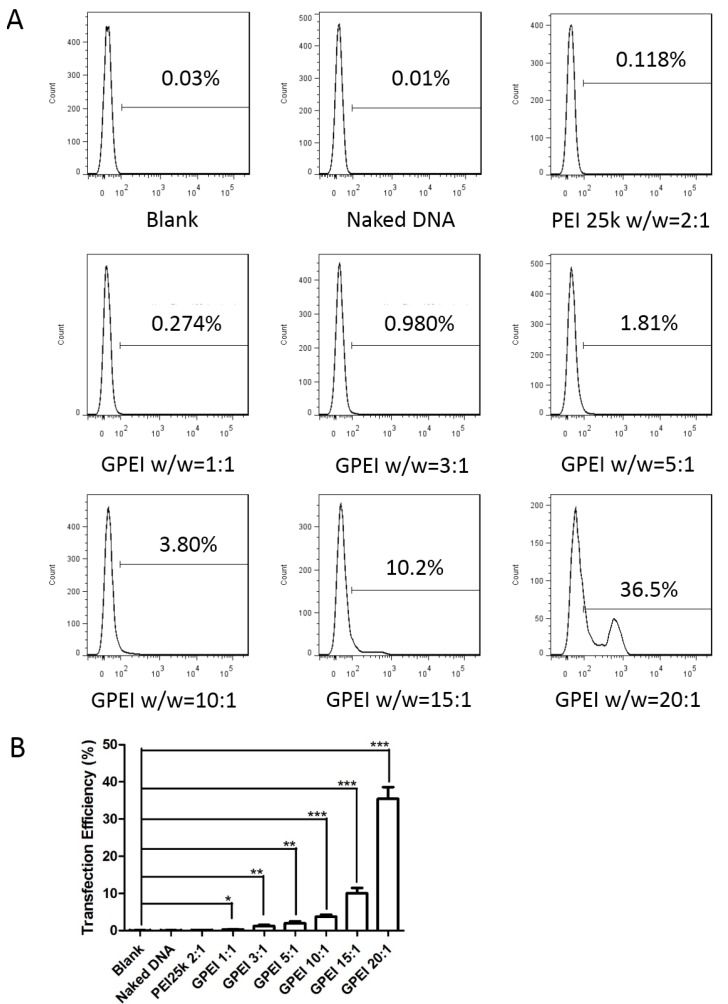
In vitro cell transfection efficiency of the GPEI polyplexes for HUVECs. (**A**) The transfection efficiency analyzed by software FlowJo. (**B**) In vitro transfection efficiency (data are presented as mean values ± S.D. (*n* = 3, *: *p* < 0.05 **: *p* < 0.01 ***: *p* < 0.001).

**Figure 7 pharmaceutics-11-00171-f007:**
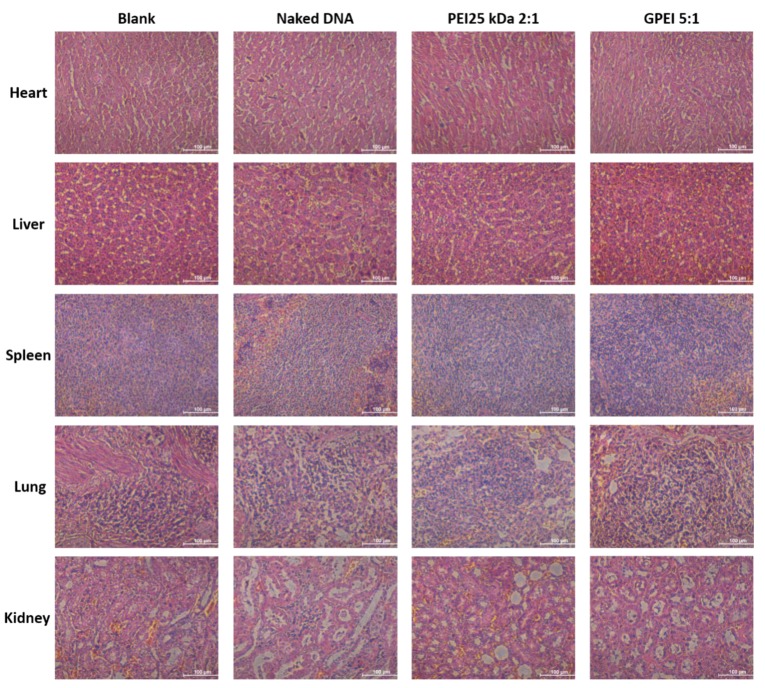
Hematoxylin–eosin (HE)-staining images of main organs. Bars = 100 μm.

**Figure 8 pharmaceutics-11-00171-f008:**
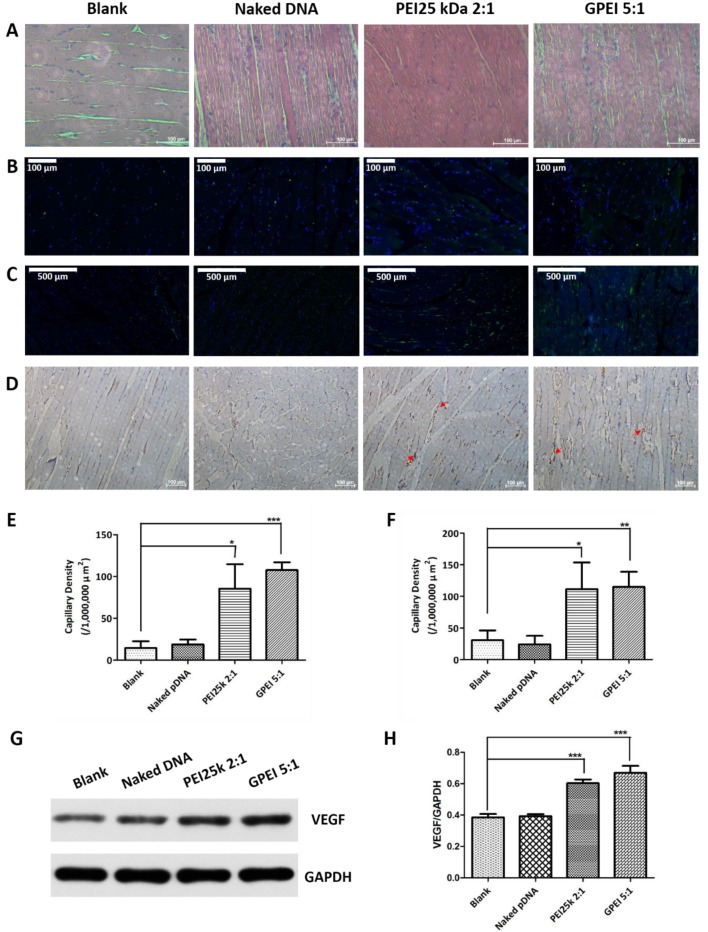
Neovascularization of gastrocnemii. (**A**) HE-staining images of gastrocnemii. Bar = 100 μm. (**B**) 5-Bromo-2-deoxyUridine (BrdU) immunofluorescence staining images of gastrocnemii (blue: DAPI; green: BrdU+). Bar = 100 μm. (**C**) CD34 immunofluorescence staining images of gastrocnemii (blue: DAPI; green: CD34+). Bar = 500 μm. (**D**) CD31 immunohistochemical staining images of gastrocnemii (brown, red arrows: CD31). Bar = 100 μm. (**E**/**F**) Image Pro Plus was used to conduct quantitative analysis of the capillary density of gastrocnemii (**E**: CD34, **F**: CD31). (**G**) The Western blotting analysis of gastrocnemius muscles. (**H**) Image J was used to conduct quantitative analysis of vascular endothelial growth factor (VEGF) levels of gastrocnemii. Data are presented as mean values ± S.D. (*n* = 4, *: *p* < 0.05, **: *p* < 0.01, ***: *p* < 0.001).

**Table 1 pharmaceutics-11-00171-t001:** Preparation of the GPEI polyplexes. PEI: polyethylenimine; GA: glutaraldehyde; pDNA: plasmid DNA.

*w*/*w* Ratio	pDNA (μL)	PEI 1.8 kDa (μL)	GA (μL)	H_2_O (μL)
1	1000	10	7	983
3	1000	30	21	949
5	1000	50	35	915
10	1000	100	70	830
15	1000	150	105	745
20	1000	200	140	660

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
