# Peer review of "A Low-Molecular-Weight Polyethylenimine/pDNA-VEGF Polyplex System Constructed in a One-Pot Manner for Hindlimb Ischemia Therapy"

_pharmaceutics, 2019, doi:10.3390/pharmaceutics11040171_

Reviewer 1 Report

In this paper, the authors claim to have made a PEI/pDNA VEGF polyplex system (GPEI) that shows lower cytotoxicity and higher transfection efficiency in vitro and in vivo. They also claim that the GPEI polyplex system can generate new capillaries in vivo by promoting blood vessel formation.

The results in figure 2 and 3 characterize the nanoparticle. Figures 4-5 show that the GPEI polyplexes are taken up by cells and do not adversely affect cell viability. Figure 6 shows GFP expression induced  by GPEI polyplexes at high GPEI : pDNA ratios. However, some results in Figure 6-8 were hard to interpret for the following reasons

1.     In in vitro experiments, GPEI 5:1 and PEI of 2:1 can only transfect ~ 0.2 and 2 % cells respectively.  In this context, it is a bit puzzling that a) these formulations were selected for in vivo experiments and b) these formulations could result in increased angiogenesis. One way to address this concern would involve  controls showing that the increased angiogenesis is due to VEGF expression from the VEGF expressing pDNA and not due to some other biological response that upregulates VEGF expression due to to the presence of polyplexes. For example, an appropriate control maybe GPEI or PEI polyplexes expressing a biologically neutral protein such as GFP. 

2.     Transfection of pDNA in primary cells is challenging due to innate immune responses. It was unclear from the methods if the HUVECs used are primary cells or an immortalized HUVEC cell line. Some discussion around these results would be appropriate.

3.     Relying on histology to claim therapeutic benefit is a bit concerning. A functional assay to measure improvement of PAD may help

Author Response

Q1. In in vitro experiments, GPEI 5:1 and PEI of 2:1 can only transfect ~ 0.2 and 2 % cells respectively.  In this context, it is a bit puzzling that a) these formulations were selected for in vivo experiments and b) these formulations could result in increased angiogenesis. One way to address this concern would involve controls showing that the increased angiogenesis is due to VEGF expression from the VEGF expressing pDNA and not due to some other biological response that upregulates VEGF expression due to to the presence of polyplexes. For example, an appropriate control maybe GPEI or PEI polyplexes expressing a biologically neutral protein such as GFP. 

RE: Thank you very much for your advances! PEI or GFP cannot upregulates VEGF expression, because the only GPEI or PEI have been confirmed that they are high toxicity and even die when they are injected in vivo blood [Pharmaceutical Research 2005,22(3):373-380. DOI: 10.1007/s11095-004-1874-y; Bioconjugate Chem. 2003, 14, 934-940; Polymer 2014, 55: 5178-5188; International Journal of Pharmaceutics 2017,521: 249–258; Molecular Pharmaceutics 2010, 7(3):727–737] and GFP polyplexes have also been confirmed that they only express GFP and do not upregulated VEGF expression [Pharmaceutical Research 2005; 22(3):373-380. DOI: 10.1007/s11095-004-1874-y; ACS Appl Mat Inter. 2018;10 (6):5196–5202].

Q2. Transfection of pDNA in primary cells is challenging due to innate immune responses. It was unclear from the methods if the HUVECs used are primary cells or an immortalized HUVEC cell line. Some discussion around these results would be appropriate.

RE: HUVECs are primary cells. Some discussion has been added in Line 95.

Q3. Relying on histology to claim therapeutic benefit is a bit concerning. A functional assay to measure improvement of PAD may help.

RE: Thank you very much for your advices! However, we measured the direct capillary density and VEGF according to the reference [Pharmaceutical Research 2005;22(7):1110-1116. DOI: 10.1007/s11095-005-5644-2]. We will measure some functional assays in the further.

Reviewer 2 Report

The authors of the manuscript titled: “Low-molecular-weight polyethylenimine/pDNA-VEGF polyplex system constructed by one-pot manner for hindlimb ischemia therapy”, developed and validated in vitro and in vivo glutaraldehyde/low-molecular-weight polyethylenimine polyplex system (GPEI) for the efficient delivery of plasmid DNA into ischemic muscle. The polyplex particles showed low toxicity and were readily uptaken by HUVEC cells. Intramuscular injections of the pDNA-VEGF containing GPEIs were able to induce vascularization of the ischemic muscle to the same or slightly higher level than much more toxic high-molecular weight polyethylenimine formulations. The simplified one-pot production and clear therapeutic effect in rat muscle injury model renders the presented work interesting, impactful and worth publishing. Unfortunately manuscript lacks proper English and requires extensive corrections of the grammar and style beyond reviewer capabilities. A few examples below:

Line 29: “PAD has gradually been increasingly popular these years” should be: The prevalence of PAD is gradually increasing in recent years due to….

Line 31: “normal people” should be: healthy subjects/individuals

Line 39: “gene drugs” should be: nucleic acid based drugs

Line 48:”naked genes” should be: naked DNA

Line 96: “and incubated” should be: and cultured

Line 191: “band appeared” should be: band present

Line 261: “can infiltrate” should be: can incorporate into

Line 294: “raising” should be: increasing

Line 298: “twice-condensation” should be: double-condensation

Additionally, the TEM picture in Fig 3 D should be replaced with a better one since the GPEI particles in this picture are heterogeneous in size opposite than postulated by authors.

Author Response

Q1.  Line 29: “PAD has gradually been increasingly popular these years” should be: The prevalence of PAD is gradually increasing in recent years due to….

Line 31: “normal people” should be: healthy subjects/individuals

Line 39: “gene drugs” should be: nucleic acid based drugs

Line 48:”naked genes” should be: naked DNA

Line 96: “and incubated” should be: and cultured

Line 191: “band appeared” should be: band present

Line 261: “can infiltrate” should be: can incorporate into

Line 294: “raising” should be: increasing

RE: Thank you very much for your help! These above errors have been modified in the manuscript.

Q2. Line 298: “twice-condensation” should be: double-condensation

RE: Thank you very much for your help! PEI 1.8k Da can condense pDNA via electrostatic interaction to form polyplexes (first condensation). And then the aldehyde groups in GA could further react with the amine groups of PEI 1.8 kDa to form imine linkages, which in theory can condense the pDNA again (second condensation). Therefore, we called it twice-condensation method.

Q3. the TEM picture in Fig 3 D should be replaced with a better one since the GPEI particles in this picture are heterogeneous in size opposite than postulated by authors.

RE: Thank you very much for your help! This question has been modified in the manuscript. (Line 204)

Reviewer 3 Report

Guo et al. used a new pDNA delivery system (GPEI) consisting of glutaraldehyde as a crosslinking agent for low molecular weight PEI. They investigated the ability to couple pDNA to the GPEI and its transfection efficiency to endothelial cells in vitro and in the hindlimb ischemia model in vivo. The authors compare the pDNA-GPEI complexes with pDNA-PEI complexes.

The main concern from my side is the fact that the advantage with GPEI versus PEI for in vivo therapeutic effects is not shown (although it is stated in the conclusion).

1)    After reading the manuscript I conclude that there is no difference in using PEI or GPEI (no difference in transfection efficiency or toxicity in vivo). The main advantage remaining is that GPEI is biodegradable (apart from its easier way of production) but this has not been shown by the authors. The manuscript would profit from an experiment showing in vitro that GPEI is degraded faster than PEI and/or that pDNA is released faster or to a higher extent from the GPEI complexes in comparison to the PEI-complexes. However, even if this is so it does not seem to make a difference for in vivo situations. So why should one rather use GPEI-pDNA instead of PEI-pDNA for therapeutic effects? This should at least be discussed thoroughly.

2)    The manuscript contains numerous language mistakes and should be checked by a native speaker.

For instance: 2nd sentence in the introduction: “PAD has gradually been increasingly popular these years…..” I presume the authors mean that the number of patients with PAD has increased the last years, not that this disease is popular.

Line 37: “While gene therapy….” This sentence is missing its subclause or it has to be rewritten without the word “While”.

Line 39: “It was reported that the preclinical study…” which study does the authors refer to (there is no reference)?

Line 42: “….which is also named as therapeutic angiogenesis.” Remove “as” and use “called” instead of “named”.

The manuscript contains further sentences like these ones.

3)    Animal experiments: Have the work been conducted in accordance to relevant national legislation on the use of animals for research? This should be stated in the Materials and Method section under Hindlimb Ischemia Model.

4)    Figure 4: State A) and B) in the legend, please.

5)    Figure 6: A quantification of several experiments should be included to show the reproducibility of the transfection efficiency.

6)    Figure 8D: You can hardly see CD31-staining in the chosen photos. Please state the colour of CD31 staining in the photos in the legend. Why was not immunofluorescence performed, as with CD34?

7)    Figure 8G: Why was not mRNA of VEGF measured? qRT-PCR offers a better method of quantification than western blots.

8)    Fig. 8F: What is actually quantified by the Pro Plus?

9)    Why was laser doppler imaging not performed in the ischemia hindlimb experiments? This is standard and confirms blood flow (functional vessels and therefore physiological response to the treatment).

10)     The discussion is missing! The results are not discussed in the result section either….

Author Response

Q1. After reading the manuscript I conclude that there is no difference in using PEI or GPEI (no difference in transfection efficiency or toxicity in vivo). The main advantage remaining is that GPEI is biodegradable (apart from its easier way of production) but this has not been shown by the authors. The manuscript would profit from an experiment showing in vitro that GPEI is degraded faster than PEI and/or that pDNA is released faster or to a higher extent from the GPEI complexes in comparison to the PEI-complexes. However, even if this is so it does not seem to make a difference for in vivo situations. So why should one rather use GPEI-pDNA instead of PEI-pDNA for therapeutic effects? This should at least be discussed thoroughly.

RE: Thank you very much for your advances! We have added some discussions. The degradation experiments of the GPEI polycation analogue at pH = 6.0 and 7.4 proved that these biodegradable linkages were pH-responsive in the follow article.  The GPEI polycation analogues had a significant molecular weight loss in formic buffer (pH = 6.0), but were hardly degradable in PBS buffer (pH = 7.4).[Biomaterials Science 2018, 6, 2059-2072.] Although the cytotoxicity of PEI25kDa polyplexes is significantly higher than that of GPEI polyplexes in vitro, since rats possibly have self-repairing function, there is no significant difference in the cytotoxicity between them in vivo for short time. However, the PEI25kDa have been confirmed that it is high cytotoxicity, because PEI25kDa is not degraded and will added in vivo cumulative toxicity for long time and some degradable PEI is easy to degraded and excreted in vitro [Pharmaceutical Research 2005,22(3):373-380. DOI: 10.1007/s11095-004-1874-y; Bioconjugate Chem. 2003, 14, 934-940; Polymer 2014, 55: 5178-5188; International Journal of Pharmaceutics 2017,521: 249–258; Molecular Pharmaceutics 2010, 7(3):727–737].

Q2. The manuscript contains numerous language mistakes and should be checked by a native speaker.

For instance: 2nd sentence in the introduction: “PAD has gradually been increasingly popular these years…..” I presume the authors mean that the number of patients with PAD has increased the last years, not that this disease is popular.

Line 37: “While gene therapy….” This sentence is missing its subclause or it has to be rewritten without the word “While”.

Line 39: “It was reported that the preclinical study…” which study does the authors refer to (there is no reference)?

Line 42: “….which is also named as therapeutic angiogenesis.” Remove “as” and use “called” instead of “named”.

The manuscript contains further sentences like these ones.

RE: Thank you very much for your help! These above errors have been modified in the manuscript.

Q3. Animal experiments: Have the work been conducted in accordance to relevant national legislation on the use of animals for research? This should be stated in the Materials and Method section under Hindlimb Ischemia Model.

RE: Thank you very much for your help! This part has been added to the manuscript. (Line 158-163)

Q4. Figure 4: State A) and B) in the legend, please.

RE: Thank you very much for your help! This part has been added to the manuscript.

Q5. Figure 6: A quantification of several experiments should be included to show the reproducibility of the transfection efficiency.

RE: Thank you very much for your help! This part has been added to the manuscript.

Q6. Figure 8D: You can hardly see CD31-staining in the chosen photos. Please state the colour of CD31 staining in the photos in the legend. Why was not immunofluorescence performed, as with CD34?

RE: Thank you very much for your help! This part has been added to the manuscript.

Tablets of immunohistochemical staining can be reviewed and store for a long time, and their positive signal amplification is higher than fluorescence. The positive signal of immunofluorescence staining is intuitive and easy to observe. Comprehensively compare the advantages and disadvantages of the two dyeing methods, we finally chose CD34 immunofluorescence staining and CD31 immunohistochemical staining.

Q7. Figure 8G: Why was not mRNA of VEGF measured? qRT-PCR offers a better method of quantification than western blots.

RE: Thank you very much for your help! mRNA needs to be translated and expressed to form VEGF protein. The content of VEGF protein can be detected by western blotting directly and the result is more intuitive.

Q8. Fig. 8F: What is actually quantified by the Pro Plus?

RE: Thank you very much for your help! Image Pro Plus was used to conduct quantitative analysis of the capillary density of gastrocnemii.

Q9. Why was laser doppler imaging not performed in the ischemia hindlimb experiments? This is standard and confirms blood flow (functional vessels and therefore physiological response to the treatment).

RE: Thank you very much for your advices! However, we measured the direct capillary density and VEGF according to the reference [Pharmaceutical Research 2005;22(7):1110-1116. DOI: 10.1007/s11095-005-5644-2]. Another, we use Image Pro Plus to quantitative analysis of the capillary density. We will measure some functional assays in the further. Our laboratory is unable to provide Doppler imaging conditions currently.

Q10. The discussion is missing! The results are not discussed in the result section either….

RE: Thank you very much for your help! This part has been added to the manuscript.Line 208-209, Line 230-236, Line 266-270, Line 272-276

Round  2

Reviewer 1 Report

The authors have addressed concern 2 from the last round of review relating to the transfection of primary cells.

It has been a persistent challenge in the field to use non-viral vectors to achieve high levels of transfection and protein production in vivo. In that sense, the in vivo results are very interesting. But the in vivo and in vitro results seem to be inconsistent with each other. The in vitro results show that only 2% of cells are transfected by GPEI. In vivo transfection efficiencies are usually significantly lower than in vitro efficiencies. In that context, the results presented here suggest that the transfection of 2% of cells (or possibly even fewer cells) with VEGF expressing plasmids is sufficient to increase VEGF expression by ~50% (Figure 8H) and increase capillary density even more dramatically (Figure 8F). It is possible that transfection of<2% cells could have such a dramatic response. But as the authors point out from the numerous references, polycations and polyplexes formed from polycations are cytotoxic both in vitro and in vivo.  Since polyplexes are cytotoxic, it is likely that repeated injection of polyplexes in the injury site causes cell death. How is this injury repaired? Does this process result in natural VEGF production? Unfortunately, the studies the authors highlighted seemed to inject polyplexes intravenously and do not assess VEGF expression or capillary formation at the injection site.  The controls presented in this study do not rule out the possibility that the increased VEGF production and capillary formation is due to the VEGF expressed from the plasmid and not due to injury induced by the polyplexes. One set of controls that can rule out this non-specific response include injections of bare polymer or injections of GPEI polyplexes containing another plasmid DNA construct that does not express a disease modifying protein. Alternately, the authors could also quantify plasmid uptake by cells in the muscle and assess if a small number of transfected cells expressing high levels of VEGF are responsible for the impressive in vivo results. If the experiments rule out the possibility of natural VEGF production in response to an injury, the results presented here would be very interesting to the field.

Author Response

Thank you very much for your letter and the comments from the referees about our paper submitted to pharmaceutics (Manuscript ID: pharmaceutics-451541)! We have checked the manuscript and revised it according to the comments. We submit here the revised manuscript as well as a list of changes.

Reviewer 1

Q1

It has been a persistent challenge in the field to use non-viral vectors to achieve high levels of transfection and protein production in vivo. In that sense, the in vivo results are very interesting. But the in vivo and in vitro results seem to be inconsistent with each other. The in vitro results show that only 2% of cells are transfected by GPEI. In vivo transfection efficiencies are usually significantly lower than in vitro efficiencies. In that context, the results presented here suggest that the transfection of 2% of cells (or possibly even fewer cells) with VEGF expressing plasmids is sufficient to increase VEGF expression by ~50% (Figure 8H) and increase capillary density even more dramatically (Figure 8F). It is possible that transfection of<2% cells could have such a dramatic response. But as the authors point out from the numerous references, polycations and polyplexes formed from polycations are cytotoxic both in vitro and in vivo. Since polyplexes are cytotoxic, it is likely that repeated injection of polyplexes in the injury site causes cell death. How is this injury repaired? Does this process result in natural VEGF production? Unfortunately, the studies the authors highlighted seemed to inject polyplexes intravenously and do not assess VEGF expression or capillary formation at the injection site.  The controls presented in this study do not rule out the possibility that the increased VEGF production and capillary formation is due to the VEGF expressed from the plasmid and not due to injury induced by the polyplexes. One set of controls that can rule out this non-specific response include injections of bare polymer or injections of GPEI polyplexes containing another plasmid DNA construct that does not express a disease modifying protein. Alternately, the authors could also quantify plasmid uptake by cells in the muscle and assess if a small number of transfected cells expressing high levels of VEGF are responsible for the impressive in vivo results. If the experiments rule out the possibility of natural VEGF production in response to an injury, the results presented here would be very interesting to the field.

RE:

Thank you very much for your help! In the experiment, polyplexes were injected intramuscularly not intravenously at the model site once every three days. The frequency of injections once every three days promoted the continued expression of VEGF protein by the cells. After the delivered VEGF plasmid entering the nucleus, the cell can express the VEGF protein slowly and continuously, so the whole process is a sustained release effect. Although the transfection efficiency detected after 72 hours was low in vitro, the cumulative expression of VEGF protein was high after long-term local injection. Moreover, the concentration of VEGF injected intramuscularly was 10 times that of the cell experiment. Therefore, the accumulation efficiency of VEGF expression in the body will be significantly higher than the cellular level in vitro. The mode of administration is intramuscular injection, and the we assessed VEGF expression and capillary formation of gastrocnemii, which was the injection site.

According to the previous work in our group [1,2] and others [3], we used polycationic materials (modified PEI 1.8 kDa) to deliver plasmid DNA encoding anti-VEGF-shRNA to inhibit VEGF expression, which could achieve tumor treatment. If injury induced by the polyplexes could promote VEGF expression, the polycation silencing system in follow articles would have no therapeutic effect. However, the experimental results showed that the plasmid DNA encoding anti-VEGF-shRNA delivered by polycationic material could reduce VEGF expression and inhibite tumor growth significantly. Polycationic materials delivering anti-VEGF-shRNA can provide a promising method for tumor treatment. Therefore, the increased VEGF production and capillary formation is due to the VEGF expressed from the plasmid and not due to injury induced by the polyplexes.

1.  Che, J.; Tao, A.; Chen, S.; Li, X.; Yi, Z.; Yuan, W. Biologically responsive carrier-mediated anti-angiogenesis shRNA delivery for tumor treatment. Sci Rep 2016, 6, 35661.

2.   Li, X.; Guo, X.; Cheng, Y.; Zhao, X.; Fang, Z.; Luo, Y.; Xia, S.; Feng, Y.; Chen, J.; Yuan, W.-E. pH-Responsive Cross-Linked Low Molecular Weight Polyethylenimine as an Efficient Gene Vector for Delivery of Plasmid DNA Encoding Anti-VEGF-shRNA for Tumor Treatment. Frontiers in Oncology 2018, 8, doi:10.3389/fonc.2018.00354.

3.  Zhou, Y.; Yu, F.; Zhang, F.; Chen, G.; Wang, K.; Sun, M.; Li, J.; Oupický, D. Cyclam-Modified PEI for Combined VEGF siRNA Silencing and CXCR4 Inhibition To Treat Metastatic Breast Cancer. Biomacromolecules 2018, 19, 392401.

       As the follow figure shown, the cytotoxicity of GPEI polymer was significantly higher than that of polyplexes. The reason for this result is that a simple polymer has more positive charges than polyplexes containing nucleic acids, which could course high cytotoxicity. Therefore, we did not use GPEI polymer to therapy in animal model.

Reviewer 3 Report

Author's Notes

The authors have addressed the reviewer´s points and changed the manuscript accordingly, which has also improved the manuscript. However, some of the points have not been addressed properly and I have some additional questions.

Q1. After reading the manuscript I conclude that there is no difference in using PEI or GPEI (no difference in transfection efficiency or toxicity in vivo). The main advantage remaining is that GPEI is biodegradable (apart from its easier way of production) but this has not been shown by the authors. The manuscript would profit from an experiment showing in vitro that GPEI is degraded faster than PEI and/or that pDNA is released faster or to a higher extent from the GPEI complexes in comparison to the PEI-complexes. However, even if this is so it does not seem to make a difference for in vivo situations. So why should one rather use GPEI-pDNA instead of PEI-pDNA for therapeutic effects? This should at least be discussed thoroughly.

RE: Thank you very much for your advances! We have added some discussions. The degradation experiments of the GPEI polycation analogue at pH = 6.0 and 7.4 proved that these biodegradable linkages were pH-responsive in the follow article. The GPEI polycation analogues had a significant molecular weight loss in formic buffer (pH = 6.0), but were hardly degradable in PBS buffer (pH = 7.4).[Biomaterials Science 2018, 6, 2059-2072.] Although the cytotoxicity of PEI25kDa polyplexes is significantly higher than that of GPEI polyplexes in vitro, since rats possibly have self-repairing function, there is no significant difference in the cytotoxicity between them in vivo for short time. However, the PEI25kDa have been confirmed that it is high cytotoxicity, because PEI25kDa is not degraded and will added in vivo cumulative toxicity for long time and some degradable PEI is easy to degraded and excreted in vitro [Pharmaceutical Research 2005,22(3):373-380. DOI: 10.1007/s11095-004-1874-y; Bioconjugate Chem. 2003, 14, 934-940; Polymer 2014, 55: 5178-5188; International Journal of Pharmaceutics 2017,521: 249–258; Molecular Pharmaceutics 2010, 7(3):727–737].

Although the authors have added some lines discussing this, it is not enough in my opinion. The results of the in vivo experiments are vital for the manuscript and the fact that no advantage of GPEI in comparison to PEI (no higher toxicity, no higher therapeutic effect) can be shown goes against the initial hypothesis of the authors. There may be good reasons or explanations for this effect (no difference) in vivo, which the authors stated above. Therefore, this has to be more extensively discussed in the manuscript. Please discuss the evidence that PEI may cause long-term toxicity while GPEI may not (if there is evidence for this) more extensively.

What do the authors mean by “self-repairing function” in rats? Please specify this and how this may be connected to the observations in vivo.

Q2. The manuscript contains numerous language mistakes and should be checked by a native speaker.

For instance: 2nd sentence in the introduction: “PAD has gradually been increasingly popular these years…..” I presume the authors mean that the number of patients with PAD has increased the last years, not that this disease is popular.

Line 37: “While gene therapy….” This sentence is missing its subclause or it has to be rewritten without the word “While”.

Line 39: “It was reported that the preclinical study…” which study does the authors refer to (there is no reference)?

Line 42: “….which is also named as therapeutic angiogenesis.” Remove “as” and use “called” instead of “named”.

The manuscript contains further sentences like these ones.

RE: Thank you very much for your help! These above errors have been modified in the manuscript.

Although the authors took great care in correcting the above comments, the manuscript still contains language errors and should be checked by a native English speaker.
For instance in the abstract: “advanced therapeutic strategy” should be advanced therapeutic strategies….. change “big challenge” to great challenge, change “owing to” to due to.

Or in the text lines 244 and 245: “…when in 24h” and “when in 4h”, please change to “after 24h” and “after 4h”

Etc.

Q5. Figure 6: A quantification of several experiments should be included to show the reproducibility of the transfection efficiency.

RE: Thank you very much for your help! This part has been added to the manuscript.

The figure has improved by the quantification. However, number of experiments is missing. Please state the n´s and run statistics. If the experiments are performed less than 3 times, please perform additional experiments.

Q6. Figure 8D: You can hardly see CD31-staining in the chosen photos. Please state the colour of CD31 staining in the photos in the legend. Why was not immunofluorescence performed, as with CD34?

RE: Thank you very much for your help! This part has been added to the manuscript.

Tablets of immunohistochemical staining can be reviewed and store for a long time, and their positive signal amplification is higher than fluorescence. The positive signal of immunofluorescence staining is intuitive and easy to observe. Comprehensively compare the advantages and disadvantages of the two dyeing methods, we finally chose CD34 immunofluorescence staining and CD31 immunohistochemical staining.

Is it possible to choose another picture, where CD-31 staining is clearer or insert arrows to point out positions with CD-31 staining?

Fig. 5: Why is the merge picture so much lighter than the single ones? Is there a transmission image in the merge? If so, please state or show this. If not, one may become the impression that the merge picture is manipulated in some way…..

Author Response

We have checked the manuscript and revised it according to the comments. We submit here the revised manuscript as well as a list of changes.

Reviewer 2

Q1

Although the authors have added some lines discussing this, it is not enough in my opinion. The results of the in vivo experiments are vital for the manuscript and the fact that no advantage of GPEI in comparison to PEI (no higher toxicity, no higher therapeutic effect) can be shown goes against the initial hypothesis of the authors. There may be good reasons or explanations for this effect (no difference) in vivo, which the authors stated above. Therefore, this has to be more extensively discussed in the manuscript. Please discuss the evidence that PEI may cause long-term toxicity while GPEI may not (if there is evidence for this) more extensively.

RE:

In the experiment, polyplexes were injected intramuscularly not intravenously at the model site, so polyplexes could not reach the body through the circulation, and it could not cause obvious toxicity to various organs. Next, the injected polyplexes are macromolecules that is not easily absorbed and distributed to various organs after local injection. When the macromolecules are degraded into small molecules, they can be absorbed and distributed, but the small molecules are low in toxicity and do not cause significant organ toxicity. In addition, PEI is associated with dose-dependent toxicity, and the toxicity of PEI is related to the molecular weight and the amount of the material. It is reported that polyplexes formed by branched PEI 25 kDa were highly toxic at N/P = 10, causing the death of all of the mice injected with them [1]. The w/w ratio of the PEI 25 kDa polyplex used in the experiment is 2, and the low proportion of PEI 25 kDa is relatively low in toxicity, so no significant toxicity has been caused. As Figure 8E-8H shown, although there were no significant differences in the results between the GPEI group and the PEI 25 kDa group, it could be seen from the figure that the treatment effect of GPEI group was better than that of PEI 25 kDa group when the two groups were compared with blank group respectively. In addition, Matthew L. Springer et al. [2] has reported that lower VEGF levels or shorter durations of exposure to VEGF induces angiogenesis in ischemic muscle, whereas higher levels or longer durations induces vasculogenesis in non-ischemic muscle. In other words, VEGF can promote angiogenesis regardless of the concentration, which may be the reason for the similar therapeutic effect in the PEI group and the GPEI group.

1.          Mini, T.; Lu, J.J.; Qing, G.; Chengcheng, Z.; Jianzhu, C.; Klibanov, A.M. Full deacylation of polyethylenimine dramatically boosts its gene delivery efficiency and specificity to mouse lung. Proc Natl Acad Sci U S A 2005, 102, 5679-5684.

2.         Springer, M.L.; Chen, A.S.; Kraft, P.E.; Bednarski, M.; Blau, H.M. VEGF gene delivery to muscle: potential role for vasculogenesis in adults. Molecular Cell 1998, 2, 549-558.

Q2

Although the authors took great care in correcting the above comments, the manuscript still contains language errors and should be checked by a native English speaker.

For instance in the abstract: “advanced therapeutic strategy” should be advanced therapeutic strategies….. change “big challenge” to great challenge, change “owing to” to due to.

Or in the text lines 244 and 245: “…when in 24h” and “when in 4h”, please change to “after 24h” and “after 4h”

RE:

Thank you very much for your help! These above errors have been modified in the manuscript.

Q3

The figure has improved by the quantification. However, number of experiments is missing. Please state the n´s and run statistics. If the experiments are performed less than 3 times, please perform additional experiments.

RE:

Thank you very much for your help! This part has been added to the manuscript.

Q4

Is it possible to choose another picture, where CD-31 staining is clearer or insert arrows to point out positions with CD-31 staining?

RE:

Thank you very much for your help! This part has been added to the manuscript.

Q5

Why is the merge picture so much lighter than the single ones? Is there a transmission image in the merge? If so, please state or show this. If not, one may become the impression that the merge picture is manipulated in some way…..

RE:

Thank you very much for your help! There is a bright filed image in the merge, so the merge picture is so much lighter than the single ones. The bright filed images of cells have been added to the manuscript.

Round  3

Reviewer 1 Report

The authors' response pointing to previous studies looking at PEI based systems to modulate VEGF expression suggest that the results presented here may not be due to the activity of free polymer. It would very useful for readers if the authors include the explanation provided in their last response with those references in the paper.

Author Response

We have checked the manuscript and revised it according to the comments. We submit here the revised manuscript as well as a list of changes.

Q: The authors' response pointing to previous studies looking at PEI based systems to modulate VEGF expression suggest that the results presented here may not be due to the activity of free polymer. It would very useful for readers if the authors include the explanation provided in their last response with those references in the paper.

RE: Thank you very much for your help! This part has been added to the manuscript (314-318).